# Effect of physicochemical parameters on the stability and activity of garlic alliinase and its use for *in-situ* allicin synthesis

Petra Janská[1], Zdeněk Knejzlík[1,2], Ayyappasamy Sudalaiyadum Perumal[3], Radek Jurok[4], Viola Tokárová[1], Dan V. Nicolau[3], František Štěpánek[1], Ondřej Kašpar[1]*

1 Department of Chemical Engineering, University of Chemistry and Technology Prague, Prague, Czech Republic, 2 Institute of Organic Chemistry and Biochemistry of the Czech Academy of Sciences, Prague, Czech Republic, 3 Department of Bioengineering, McGill University, Montreal, Quebec, Canada, 4 Department of Organic Chemistry, University of Chemistry and Technology Prague, Prague, Czech Republic

* kasparo@vscht.cz

**Data Availability Statement:** All relevant data are within the paper.

**Funding:** - Technology Agency of the Czech Republic (https://www.tacr.cz/en/) - grant no.

## Abstract

Garlic is a well-known example of natural self-defence system consisting of an inactive substrate (alliin) and enzyme (alliinase) which, when combined, produce highly antimicrobial allicin. Increase of alliinase stability and its activity are of paramount importance in various applications relying on its use for *in-situ* synthesis of allicin or its analogues, e.g., pulmonary drug delivery, treatment of superficial injuries, or urease inhibitors in fertilizers. Here, we discuss the effect of temperature, pH, buffers, salts, and additives, i.e. antioxidants, chelating agents, reducing agents and cosolvents, on the stability and the activity of alliinase extracted from garlic. The effects of the storage temperature and relative humidity on the stability of lyophilized alliinase was demonstrated. A combination of the short half-life, high reactivity and non-specificity to particular proteins are reasons most bacteria cannot deal with allicin's mode of action and develop effective defence mechanism, which could be the key to sustainable drug design addressing serious problems with escalating emergence of multidrug-resistant (MDR) bacterial strains.

## Introduction

Rising awareness about sustainability and associated development of green chemistry lead to the growing popularity and demands for commercial products with minimal environmental impact, with present emphasis on the origin of material resources, bioaccumulation, biodegradability, and greenhouse gas and waste production [1]. Sustainable chemistry has become a key focus area for chemical companies [2], which motivates pharmaceutical, cosmetic and food industry to employ more extensively biocatalysis as an environmentally friendly and effective alternative to traditional synthetic preparation [3–6]. Additionally, green chemistry provides an excellent opportunity for the demonstration of the important benefits of modern

TJ01000313, receivers: VT, OK, PJ - Canadian Natural Sciences and Engineering Research Council (https://www.nserc-crsng.gc.ca/index_eng.asp) - grant no. RGPIN-2016-05019, receivers: DVN, ASP - Specific university research grants of the University of Chemistry and Technology in Prague (https://www.vscht.cz/?jazyk=en) - grant No. A1_FCHI_2020_004 and A1_FCHI_2020_005, receivers: PJ, FS, VT, OK.

**Competing interests:** The authors have declared that no competing interests exist.

chemistry, often associated with disastrous poisoning accidents, global pollution, and exploitation of natural resources.

During the past decade, enzymes have proved to be an economical and sufficiently stable replacement for various chemical catalysts used commercially for both low-cost bulk products (ethanol by fermenting sugars), and for fine chemicals with high added value and purity (enantiomerically pure chiral compounds, optically active pharmaceuticals, plant protecting agents, and fragrances) [7]. Since massive commercialization in the 1990s, enzymes have been used extensively in various daily applications such as waste treatment (degradation of plastic, hydrolysis of lignocellulose), cleaning or bioremediation. However, industrial applications bring additional requirements on enzyme functionality due to environmental conditions significantly different than their natural sources, such as the presence of artificial substrates, organic solvents, inhibitors, non-optimal pH, elevated temperature, high ionic strength, which could result in both premature enzyme degradation, and loss of enzyme activity [8]. Therefore, comprehensive knowledge about enzyme long-term stability under various conditions is crucial for the effective enzyme use in the free or immobilized form [5, 9]. Since enzymes are prone to lose a significant part of their activity before use, long-term storage stability is an additional critical factor that must be considered and addressed during the development of new products [4, 5, 9–12].

Alliinase (EC 4.4.1.4), an enzyme found in garlic and other plants of genus *Allium* in exceptionally high concentrations [13], is responsible for the formation of bioactive compound allicin from stable precursor alliin. This reaction serves as a self-defence mechanism of *Allium* plants initiated when a cellular structure is compromised, and the initially separated enzyme and the substrate are mixed. A combination of the short half-life, high reactivity and non-specificity to particular proteins [14] is the reason most bacteria, and other pathogens cannot deal with allicin mode of action and develop effective defence mechanisms. It has been reported that the development of bacterial resistance against potent, but highly reactive and unstable allicin is more than 1000 times slower compared to other antibiotics [15]. In order to harness the enzymatic production of allicin or its analogues on an industrial scale without significant loss of enzyme activity [16–20], its response to various environmental conditions accompanying enzyme extraction, purification, and storage must be examined and comprehensively detailed.

It has been shown by Weiner et al. that alliinase monomer contains 10 cysteine residues, of which eight form four S-S bridges responsible for the properly folded active state, and two are present in free thiol groups [21]. Purified alliinase stored in buffer solution without the presence of any antioxidizing agents is prone to air oxidation of cysteine, followed by the formation of inactive S-S bridged alliinase clusters consisting of four, or more enzyme subunits [22]. There are several commonly used substances which can affect enzymatic behaviour in different ways depending on the nature of target protein, i.e. positively or negatively. In general, additives such as salts, polyols, sugars or inert polymers, such as polyethylene glycol, can greatly enhance the stability without the need of chemical or genetic manipulation of the target protein [23]. Chhabria et al. reported purification of alliinase from soil bacterium *Cupriavidus necator* and found that the alliinase activity is enhanced by several metal ions, e.g., $Ca^{2+}$, $Mg^{2+}$, $Zn^{2+}$, $Fe^{2+}$, $Mn^{2+}$), but inhibited by others, e.g., $Cu^{2+}$, $Cs^+$, $Li^+$, or $Hg^{2+}$ [24]. It was also shown that the addition of $K^+$, $Na^+$, and EDTA had no significant effect on the alliinase activity. Moreover, the effect of additives for alliinases of different origin may vary significantly. For example, it was reported that alliinase extracted from garlic is stimulated in the presence of EDTA [25]. The positive effects of glycerol, sucrose and sorbitol, osmolytes known as chemical chaperones, and mannose-specific lectin (Allium sativum agglutinins I) on alliinase stabilization were recently presented by Shin et al. [26].

The present study provides a comprehensive insight into the alliinase stability and activity in the solution and lyophilized state. The influence of commonly used buffers, additives, i.e., chloride salts, such as NaCl, KCl, CaCl₂, and MgCl₂, antioxidants (ascorbic acid and DTT), polyols and their combinations in various concentrations, on the enzyme activity and stability over an extended period of time (resistance to denaturation and oxidation), were investigated and discussed. The long-term stability of the enzyme in a lyophilized form stored under controlled conditions, i.e. oxygen presence, temperature and humidity, was investigated providing valuable insight into the problematics of enzyme use in dry powder formulations. Finally, antibacterial activity upon addition of substrate against Gram-positive (*S. epidermidis*) and Gram-negative (*E. coli* and *P. putida*) bacterial strains was investigated using static and dynamic *in vitro* assays.

## Materials and methods

### Chemicals

The following chemicals were purchased from Sigma-Aldrich: (±)-L-alliin (primary reference standard), Ethylenediaminetetraacetic acid (EDTA), Pyridoxal 5′-phosphate hydrate (PLP), Dithiothreitol (DTT), N[Tris(hydroxymethyl)methyl]glycine (Tricine), Polyethylene glycol (PEG) 6000, Bovine serum albumin (BSA), Nutrient Broth (NB), Kanamycin sulfate, Fluorescein diacetate, Propidium iodide (PI), Polyethylenimine (PEI). Glycerol, Sodium phosphate dibasic heptahydrate, Sodium phosphate monobasic decahydrate, Potassium dihydrogen phosphate, Phosphate-buffered saline (phosphate buffer, potassium chloride and sodium chloride), citric acid, hydrochloric acid, ascorbic acid, tryptone, yeast extract, sodium chloride, potassium chloride, calcium chloride, magnesium chloride, lithium chloride, and potassium hydroxide were purchased from Penta (Czech Republic). Magnesium nitrate hexahydrate was purchased from Fisher Scientific. L-Lactate dehydrogenase (LDH) from rabbit muscle and β-Nicotinamide adenine dinucleotide (NADH) disodium salt grade II (approx. 98%) were purchased from Roche Diagnostic (Germany). Garlic was purchased from Farmers' Market, Zahradnictví Riegelová, country of origin Czech Republic.

### Alliin synthesis

Alliin was synthesized by a method previously published by Stoll and Seebeck [27]. In the first step, L-cysteine (50 g) was dissolved in an aqueous solution of ammonium hydroxide (2M, 1200 mL) and alkylated with allyl bromide (75.0 g) at 0˚C. After 40 minutes, the reaction was terminated, and volume reduced using a rotary evaporator. The raw product (L-deoxyalliin) was filtered, washed with ethanol, vacuum dried and recrystallized from 2:3 water/ethanol mixture. In the second step, L-deoxyalliin (53.0 g) was suspended in 530 mL of water at 25˚C and mixed with 37 mL of hydrogen peroxide (30% w/w). After two days, the volume was reduced by rotary evaporator, 400 mL of acetone was added, and the mixture was stirred for 2 hours. Then, the crude (±)-L-alliin was filtered, repeatedly washed with a mixture of acetone/water (5:1) and vacuum dried. The purity of synthetic alliin (≥ 90%) was determined by HPLC (Agilent 1260 Infinity) using commercially available (±)-L-alliin as standard.

### Extraction and purification of alliinase

The crude alliinase was obtained from fresh garlic stored at cold and dark prior to its use. All operations regarding enzyme extraction were carried out at 4˚C, unless stated otherwise. The isolation protocol was followed, as described by Mallika with some minor modifications [28]. The garlic was peeled and homogenized by hand mixer in the ratio of 1:1.5 (w/v) in buffer

(pH 6.5, 20 mM sodium phosphate buffer, 10% (v/v) glycerol, 5 mM EDTA, 5% (w/v) NaCl and 20 μM PLP). The presence of PLP (cofactor of alliinase), EDTA (chelating agent of heavy metals) and glycerol (inhibitor of enzyme unfolding and aggregation) is recommended to stabilize the enzyme during extraction steps [25]. The garlic slurry was squeezed through four layers of cheese cloth followed by vacuum filtration via 5 μm pore glass filter. The fractional precipitation of proteins was performed with PEG 6000 (25% w/w), followed by centrifugation at 30,000 g for 30 min. Next, the yellowish pellet was collected and resuspended in 20 mL of deionized water with 20 μM PLP. The solution was centrifuged again at 30,000 g for 30 min, and the supernatant was passed through at 0.45 μm syringe PVDF filter to remove residual tissue fragments. Finally, the purified solution was quickly frozen in liquid nitrogen and lyophilized (Sentry 2.0, SP Scientific, USA) at -90˚C under chamber pressure of 1 μbar for 4 days. The dry product was stored in a freezer at -20˚C prior to use.

## Characterization of alliinase

The characterization of the enzyme was carried out by using sodium dodecyl sulfate-polyacrylamide gel electrophoresis (SDS-PAGE) under reducing conditions (in the presence of β-ME). SDS-PAGE was carried out using 10% (w/v) polyacrylamide gel at a constant voltage of 20 V/cm of gel length, in which broad-range molecular mass standards (the protein marker VI (10–245), AppliChem, USA) were run simultaneously. The protein content was measured by the UV/VIS spectrophotometry at $\lambda = 595$ nm, according to the Bradford method using BSA (0 to 10 μg protein/mL) as an external standard [29].

## Enzyme assay

The specific activity of the enzyme alliinase was determined via coupled NADH-dependent pyruvic acid reduction in the presence of lactate dehydrogenase (LDH) [30]. Pyruvic acid, released as a co-product during allicin formation (Fig 1), was reduced, in the presence of LDH, to lactic acid and NADH oxidized to $NAD^+$. The time-dependent disappearance of the NADH was measured by the UV/VIS spectrophotometry at $\lambda = 340$ nm. The decrease in absorbance was proportional to the reaction rate and thus to the enzyme activity. The typical alliinase assay solution (0.1 mL) contained 20 mM alliin, 20 μM PLP, 200 mM Tricine-KOH buffer (pH 8.0), 0.8 mM NADH, LDH (2.5 μL, 550 units/mg) and lyophilised alliinase at indicated concentrations. Both NADH, LDH and alliin were added in excess to ensure alliin conversion was the rate-limiting step of the assay.

The Michaelis–Menten kinetic constants, i.e., the maximum velocity ($V_{max}$) and Michaelis–Menten constant ($K_m$) of the purified alliinase in respect to the mixture of alliin diastereomers, were determined experimentally using alliin initial concentration in the range of 0.5 to 50.0 mM. Kinetic constants of Michaelis-Menten equation ($V_{max}$, $K_m$) were calculated by non-linear least-square regression using software SigmaPlot 11 (Systat Software Inc., Chicago, IL).

## Effect of temperature and pH on alliinase activity

The influence of temperature on the specific activity of purified alliinase incubated in Tricine-KOH buffer (200 mM, pH 8) was studied in detail for the range of temperatures 20˚C to 42˚C using a mixture of alliin diastereomers (20 mM) as a substrate.

The time-dependent stability of solubilized alliinase was studied at various temperatures covering the typical range for the storage and use, i.e., -20˚C, 4˚C, 25˚C and 37˚C. The samples stored at -20˚C were unfrozen before each experiment. Sodium azide (0.02% w/v) was used as an antimicrobial agent to prevent bacterial contamination during the long-term incubation.

**Fig 1. A) enzymatic formation of allicin from alliin; B) the coupled reaction of pyruvic acid, a co-product of reaction alliin with alliinase, with NADH and LDH used for UV-VIS spectroscopic assay.**

The effect of pH on alliinase activity was determined by the incubation of the enzyme in a series of buffers covering a pH range from 1 to 8 at 25°C for 10 min. The alliinase activity was determined by enzyme assay performed at a given pH. Hydrochloric acid-potassium chloride (50 mM, pH 1 to 2), citrate-phosphate (50 mM, pH 3 to 5) and sodium phosphate (100 mM, pH 6 to 8) buffers were used, and their effect was determined. The highest observed enzyme activity was taken as the reference value of 100%, and the other experimental datapoints were rescaled accordingly. The relative expression of enzyme activity was preferred where possible, since it allowed the comparison of enzymes prepared in multiple batches, where the absolute value of enzyme activity may vary depending on garlic origin, enzyme content and purity of extracted alliinase.

The influence of various buffers of the same pH on alliinase activity was studied using sodium phosphate (100 mM, pH 6–8), potassium phosphate (100 mM, pH 6–8), phosphate-buffered saline (200 mM, pH 7.4) and Tricine–KOH (200 mM, pH 8) buffers.

### Effect of additives on alliinase stability

The effects of chloride salts (NaCl, KCl, $CaCl_2$, $MgCl_2$) at different concentrations (5 mM, 50 mM, 100 mM and 500 mM) and stabilizers (ascorbic acid (4 mM), EDTA (100 mM), DTT (5 mM), glycerol (0.1 to 10 v/v%)) on alliinase stability in Tricine-KOH (200 mM, pH 8) buffer were investigated for incubation times up to 24 hours at 25°C. The activity of the purified alliinase without any additives was taken as a reference value, i.e., 100%.

### Stability of lyophilized enzyme

The sample of the enzyme (lyophilized powder, 0.05 g) in 1.5 ml polypropylene microtube was placed upright in a glass beaker. The beaker was kept sealed at 25°C or 37°C in the thermostatic oven to ensure constant conditions. For an inert atmosphere, the beaker was placed into a desiccator, and nitrogen gas was repeatedly passed through to purge all oxygen and water vapours. The saturated salt solution (LiCl, $Mg(NO_3)_2$ and NaCl) was used to maintain the

**Table 1. Conditions for testing of enzyme stability at 25˚C and 37˚C.**

| Temperature [˚C] | Relative humidity [%] | | | | |
|---|---|---|---|---|---|
| | N₂ | LiCl | Mg(NO₃)₂ | NaCl | water |
| 25 | 0 | 11.30 ± 0.27 | 52.89 ± 0.22 | 75.29 ± 0.12 | 100 |
| 37 | 0 | 11.23 ± 0.21 | 49.17 ± 0.47 | 74.77 ± 0.13 | 100 |

desired relative humidity [31] with respect to the theoretical values shown in Table 1. A sample of the powdered enzyme was analyzed for the remaining enzyme activity every 24 hours.

## Antibacterial assays

*Escherichia coli* K12 (EC43) and *Pseudomonas putida* were chosen as model Gram-negative bacteria and *Staphylococcus epidermidis* as a Gram-positive bacterium. Bacterial strains were kindly provided by Dr Zdeněk Knejzlík (Academy of Science of the Czech Republic, *E.coli* K12 DSMZ-German Collection of Microorganisms and Cell Cultures GmbH) and prof. Dan V. Nicolau (Department of Bioengineering, McGill University, *P. putida*–ATTC, Manassas, USA; *S. epidermidis*—Carolina Biological Supply Company, Burlington, USA). All bacterial strains are classified as low biological risk organisms (BSL-1).

**Cultivation of bacteria.** The cultivation of Gram-negative bacteria (*E. coli* and *P. putida*) at 37˚C used Luria-Bertani medium (LB). LB medium was prepared by dissolution of 20 g of LB salt mixture (10 g Tryptone/Peptone, 5 g yeast extract, 5 g NaCl) in 1000 mL of deionized H₂O adjusted to pH 7.2–7.4 with 1 M NaOH. For the preparation of a solid medium, 1.5 g of agar per 100 mL of liquid LB medium was added. *S. epidermidis* was cultivated at 30˚C in nutrient broth (NB) medium (8 g/L). For the preparation of a solid NB medium, 1.5 g of agar per 100 mL of liquid NB medium was added.

**Disk diffusion assay.** The antimicrobial effect of allicin was measured by a disk diffusion assay on an agar plate inoculated by bacterial suspension (OD₆₀₀ = 0.2). Different concentration of enzyme solution (0.02 mg/mL up to 10 mg/mL) was prepared using Tricine-KOH buffer (200 mM, pH 8). The mixture of the enzyme (10 μL) and substrate (10 μL, 100 mM) was added onto filter paper discs (6 mm in diameter) placed on the agar plates incubated upside down at 37˚C for 24 hours. The antibiotic kanamycin (50 mg/mL, 20 μL) was used as a positive control. A solution of pure alliin (labelled as S, 20 μL, 100 mM) or alliinase (labelled as E, 20 μL, 10 mg/mL) was used as a negative control. The sensitivity of bacterial strains to formed allicin was determined by measuring the diameter of inhibition zones using ImageJ software [32]. All experiments were performed in triplicates for each concentration. If the inhibition zones were absent, the sample was regarded as ineffective against the respective bacterial strain. The inhibition zone for kanamycin positive control was taken as a reference value, i.e. 100%.

**Cell viability analysis.** *Live/Dead assay*. Confocal laser scanning microscopy was used to image the proportions of live and dead cells using Live/Dead assay kit (Sigma-Aldrich) based on the *in vivo* staining by fluorescein diacetate (FDA) and propidium iodide (PI) fluorescent dyes. The green fluorescent FDA dye permeated the intact membrane of the cells binding to nucleic acids, whereas the red fluorescent PI dye can enter only the non-viable cells with a damaged cell membrane. The bacterial suspension (300 μL) stained according to the manufacturer protocol (1 μL of FDA with 1 μL of PI per 1 mL of bacterial suspension) was incubated with the reaction mixture of alliin (100 μL, 100 mM) and alliinase (100 μL, 1 mg/mL). Aliquots withdrawn at the specific times (0, 10, 20 and 30 min) were examined by inverted confocal microscope IX83 (Olympus), and the images were analyzed using ImageJ software. The green

and red channels were split, converted to 8-bit colour depth, thresholded and binarised. The particle count corresponded to the total number of cells stained with green, and red fluorescent dyes, respectively. The survival fraction was calculated as the ratio of viable cells vs overall cell count. All samples were plated in a triplicate under sterile conditions.

*Plate count method*. The bacterial suspension (300 μL) was incubated with the reaction mixture of alliin (100 μL, 100 mM) and alliinase (100 μL, 1 mg/mL). Samples of bacterial suspension (10 μL) with, or without alliin/alliinase were taken at the specific times (0, 10, 20 and 30 min) followed by 1000x dilution with cultivation medium (LB or NB) to suppress allicin effect. A prepared bacterial suspension was plated on Petri dishes, and a total count of emerging colonies after 24h incubation was monitored using ImageJ.

**Analysis of bacterial morphology.** The effect of *in-situ* formed allicin on bacterial morphology was examined by FEI Quanta 450 Environmental Scanning Electron Microscope (FE-ESEM). A glass coverslip serving as bacterial support was cleaned with acetone and isopropanol, activated by oxygen plasma (Harrick Plasma), and incubated in 1 w/v % PEI solution to enhance bacterial adhesion. The modified coverslips placed separately in a 6-well plate were incubated for 5 min in bacterial culture ($OD_{600}$ = 0.2), followed by addition of reaction mixture of alliin (100 μL, 100 mM) and alliinase (100 μL, 1 mg/mL). The coverslips withdrawn at the specific times (0, 10, 20 and 30 min) were immediately dehydrated with incrementally graded ethanol series (starting with 20 v/v % to anhydrous ethanol). Solvent dehydration followed by critical point drying (CPD, Leica EM CPD300) ensured the preservation of delicate sample morphology. The dehydrated samples were coated, prior SEM measurements, with a platinum layer to increase the conductivity of sample and SEM image clarity.

## Statistical evaluation

Enzyme activities were expressed as means of 6 or more measurements and their respective standard deviation (SD). Data were analyzed using one-way ANOVA followed by Tukey and Dunnett tests to compare the influence of particular additives on enzyme activity. Values of P < 0.01 were considered to be statistically significant. Statistical analyses were performed using software SigmaPlot 11.

## Results

### Characterization of alliinase

The purified garlic alliinase, analyzed by 10% reducing SDS-PAGE showed a 51 kDa single band corresponding to alliinase monomer (Fig 2A), which is in agreement with previously published reports [33–35]. SDS-PAGE analysis revealed that alliinase obtained by PEG precipitation was of high purity without a significant presence of the accompanying proteins or degradation products. The lyophilized alliinase was found to have a protein content of 88 ± 4% per weight, accompanied by non-protein substances (polysaccharides, salts, PEG) and moisture. Kinetic parameters $V_{max}$, $K_m$ and turnover number $k_{cat}$ for alliinase following Michaelis-Menten kinetics with respect to the mixture of alliin diastereomers were calculated as 18.9 ± 0.3 mM·s$^{-1}$·mg$^{-1}$, 4.45 ± 0.36 mM and 193 s$^{-1}$, respectively (Fig 2B). The calculated values of kinetic parameters correspond to previously reported data [36].

### Effect of pH, temperature and buffer on alliinase stability and activity

In general, both pH and temperature are significant factors governing enzyme activity and stability. The alliinase activity was tested for buffers from pH 1 to 8 and a narrow pH optimum at 7.0 was observed (Fig 3A), which corresponds to results reported in the literature [24, 37].

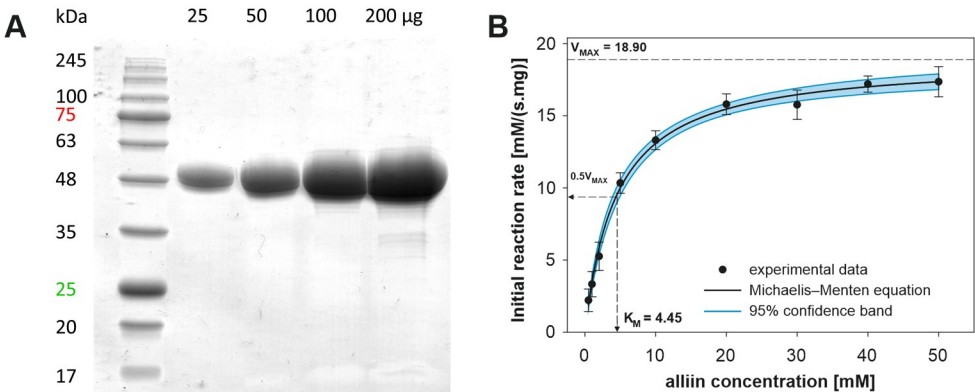

**Fig 2. A) 10% SDS-PAGE of isolated garlic alliinase; B) kinetic analysis of alliinase catalyzed reaction.** The reaction mixture (0.1 mL) contained alliinase (5 μg/mL) and various concentrations of alliin (0 to 50 mM), the reaction temperature was 25°C, Tricine-KOH (pH 8) buffer was used for the assays.

Bellow pH 5.0 and above pH 8.0, a sharp decrease in the enzyme activity was observed [38, 39]. LDH concentration in the reaction mixture was chosen such that the conversion of pyruvic acid was significantly faster compared to its formation. Moreover, the activity of LDH under various conditions was examined separately to prevent underestimation of alliinase activity caused by the slower conversion of pyruvic acid and NADH oxidation in the second reaction step. LDH has proven to be stable over a broad range of buffer composition, temperature and pH. It should be noted that under acidic conditions (pH ≤ 4) both instability of NADH and lower LDH activity may affect data interpretation, although the measured activity of alliinase at pH 4 was already negligible.

In previously published studies, various buffers were used for alliinase extraction, purification and storage, e.g., Na-PB [22, 24, 34], K-PB [30], Na/K-PB [40, 41], PBS [21, 24] and Tricine [42–44]. However, the character of various buffers may play different roles concerning the conformational, colloidal and interfacial stability of enzyme [45]. Here, a direct comparison of the most used buffers and their respective role in alliinase activity is demonstrated. The results for sodium (Na-PB) and potassium (K-PB) phosphate buffers (pH 6 to 8), PBS (pH 7.4) and Tricine-KOH (pH 8) used as the reference are shown in Fig 3B. Enzyme activity in both phosphate buffers at pH 6 was comparable. Enzyme incubated in K-PB at pH 7 showed 11%

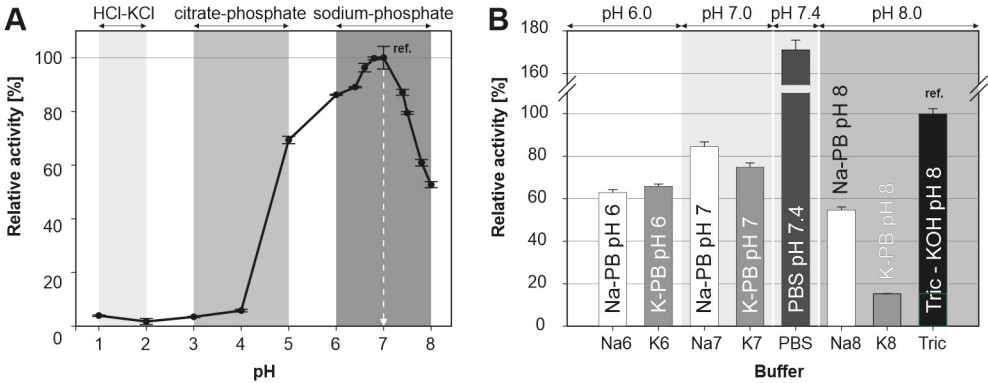

**Fig 3. Effect of pH (A) and type of used buffer (B) on the alliinase activity at 25°C.**

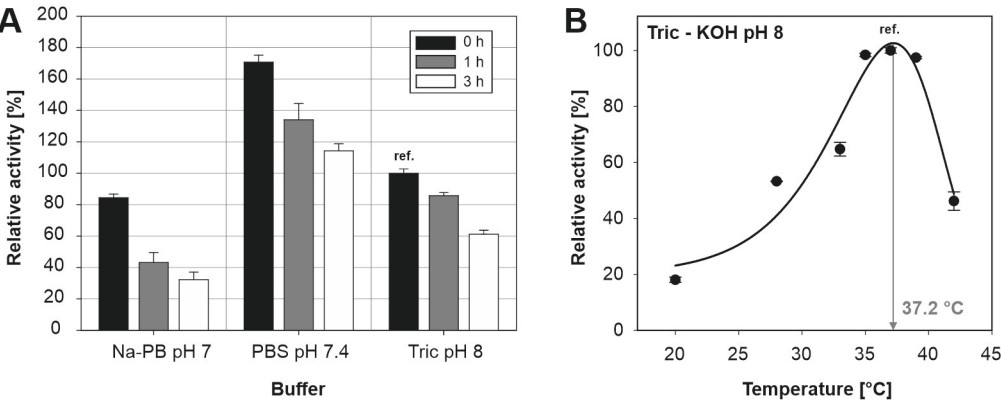

**Fig 4.** A) Time-dependent enzyme stability and activity of alliinase in Na-PB, PBS and Tricine-KOH buffer. All results are normalized to alliinase activity measured at 0 h in Tricine-KOH buffer; B) effect of temperature on alliinase activity in Tricine-KOH buffer.

lower enzyme activity compared to Na-PB. This difference was even more evident at pH 8, as the enzyme activity in K-PB was compared to Na-PB and Tricine-KOH more than 3.6-fold and 6.5-fold lower, respectively. The highest measured activity for all investigated buffers, i.e., 1.7-times higher compared to Tricine-KOH, was observed for PBS (pH 7.4), which can be explained by different pH and presence of dissolved monocations $Na^+$ and $K^+$.

Time dependant enzyme activity was studied for Na-PB, PBS and Tricine-KOH at 25°C (Fig 4A). Alliinase was pre-incubated in a corresponding buffer (Na-PB, PBS and Tricine-KOH) for three hours, and the alliinase activity was measured at specified times and the identical conditions. The alliinase activity decreased in all tested buffers with time. After 3 hours, PBS and Tricine-KOH buffer preserved 67% and 61% of the initial enzyme activity, respectively, whereas alliinase stored in the Na-PH buffer showed only 38% of its initial activity.

The alliinase activity was also studied at various temperatures within the range 20°C to 42°C in Tricine-KOH buffer (Fig 4B). Alliinase and alliin, pre-incubated separately for 5 min at the desired temperature, were mixed, and alliinase activity was measured. The bell-shaped dependence of enzyme activity on temperature showed the highest activity between 35°C to 40°C. Alliinase activity decreased rapidly over 40°C, which is in agreement with the findings reported elsewhere [46–48]. Nevertheless, for various bacterial alliinases, it has been shown that optimal temperature could be slightly shifted. Yutani et al. found that for alliinase extracted from *Ensifer adhaerens* the optimal temperature was 30°C [49]. In the case of immobilized alliinases optimum shifts to the higher temperatures, which is given by higher thermal stability of immobilized enzyme compared to its dissolved form [37].

Time-dependent alliinase stability in Tricine-KOH buffer at -20°C, 4°C, 25°C and 37°C is summarised in Table 2. It was observed that the activity of samples stored at -20°C decreased after two days by 32%. This significant decrease was most probably caused by repeated freeze-thaw cycles before every enzyme assay. The samples stored in the fridge at 4°C and 25°C

**Table 2. Effect of time and storage temperature on alliinase activity in Tricine-KOH buffer.**

| temperature [°C] | -20 | 4 | 25 | 37 |
|---|---|---|---|---|
| day 1 | 90.5 ± 2.5 | 36.3 ± 0.4 | 10.4 ± 1.5 | 5.8 ± 0.1 |
| day 2 | 67.7 ± 4.4 | 16.0 ± 2.2 | 3.6 ± 2.6 | 1.8 ± 1.3 |

Note: 100% corresponds to the initial enzyme activity at 25°C.

showed only 16% and 4% of the initial activity. Finally, the dissolved enzyme stored at 37°C was almost completely deactivated. Based on these findings, it can be concluded that alliinase in a solution cannot be effectively stored for a long time, unless frozen or without the presence of additional stabilizers, as discussed further.

### Influence of additives on enzyme activity—Operational stability

**Effect of chlorides of mono- and divalent cations.**   Enzymes can be stabilized by increasing their concentration in the solution and by modulation of the ionic strength [4]. Several salts are frequently used to enhance the stability of the enzymes, but not all are equally effective in the stabilization of a specific protein. Therefore, the effect of NaCl, KCl, $CaCl_2$ and $MgCl_2$ on the alliinase activity was investigated in 5, 50, 100 and 500 mM concentrations in Tricine-KOH buffer. All salts except $MgCl_2$ proved to have a noticeable positive effect on enzyme stability after 3 hours even at lowest concentrations (5 mM) compared to pure Tricine-KOH buffer without any additives (Fig 5). The most noticeable effect was observed for $Na^+$ (50 to 500 mM) and $Mg^{2+}$ cations (100 and 500 mM) resulting in substantially higher enzyme activity after 3 hours compared to initial activity in pure Tricine-KOH. The considerable increase of activity can be attributed to the vital role of dissolved salts in the stabilization of enzyme conformation by neutralization of protein charges or formation of salt bridges by divalent cations or to the fact that these ions can offer protection to thiols or other functional groups against oxidation by salting-out dissolved oxygen [4]. Increasing the concentration of mono or divalent cations from 100 mM to 500 mM did not show any further enhancement in terms of the alliinase activity.

**Effect of additives on alliinase stability.**   In most cases, the actual mechanism of stabilization of enzymes by altering their surrounding microenvironment is difficult to predict because of its complexity. Therefore, a complete acquaintance of additives with respect to alliinase is needed in order to assess their stabilization effect and role played in the retention of enzyme activity. The effects of ascorbic acid (antioxidant and reducing agent), glycerol (cosolvent), EDTA (chelating agent) and DTT (a strong reducing agent preventing oxidation of thiol groups and formation of disulphides) were investigated and discussed (Fig 6A).

Ascorbic acid, which acts as a radical scavenger, and donor and acceptor in electron transfer reactions [50] increased enzyme activity significantly. The presence of ascorbic acid in the

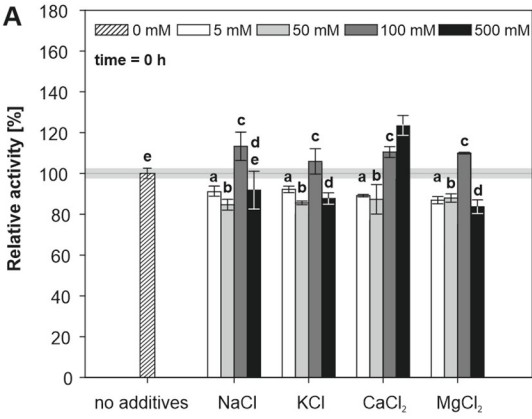
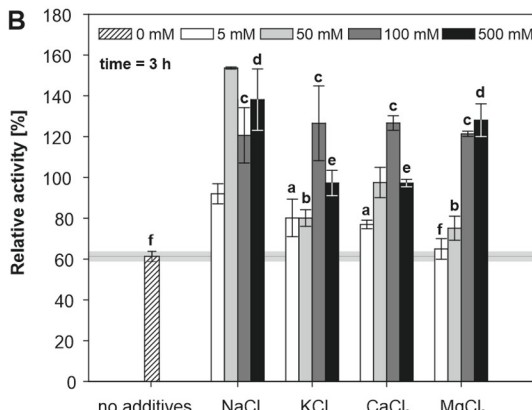

**Fig 5. Effect of salts on the initial activity of alliinase (A) and 3 h (B) incubation at 25°C.** All results are normalized to alliinase activity measured at 0 h (25°C, Tricine-KOH buffer, no additives). Means followed by the same letter (a–f) were not significantly different (P>0.01, ANOVA, Tukey's test).

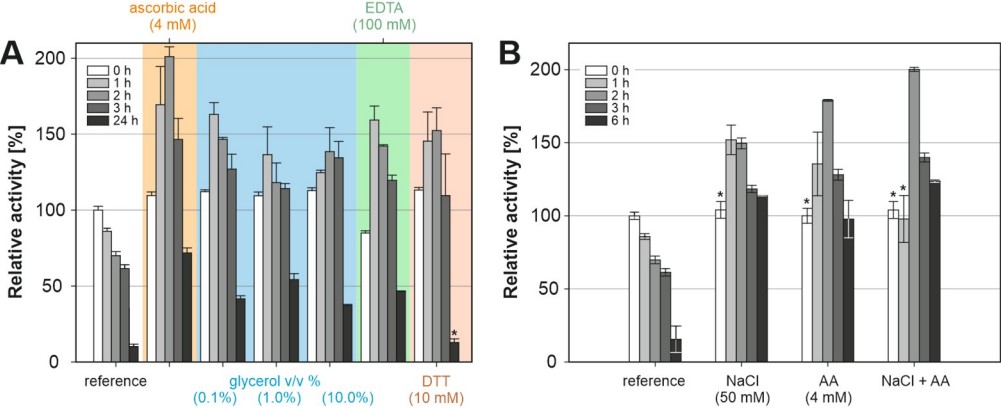

**Fig 6. A) The effect of additives on enzyme stability at 25°C; B) comparison of the individual and combined effect of 4 mM ascorbic acid (AA) and 50 mM NaCl on enzyme stability.** All results are normalized to alliinase activity measured at 0 h (25°C, Tricine-KOH buffer) without additives; * results did not differ statistically from the reference at given time (P>0.01, ANOVA, Dunnett's test).

concentration 4 mM resulted in 2.9-fold and 7.2-fold increase in enzyme activity compared to reference Tricine-KOH buffer, after 3 and 24 hours at 25°C, respectively.

It has been reported that the addition of polyols to an aqueous solution of enzyme improves strengthening of the hydrophobic interactions among non-polar amino acid residues, resulting in more compact protein conformations [51]. The addition of glycerol (0.1, 1.0, 10.0% v/v) was tested to assess the stabilization of alliinase. However, no relationship between the overall glycerol content and observed alliinase activity has been found. In all cases, the presence of glycerol preserved the activity of alliinase for 3 hours, which was in all cases higher than the initial reference value. After one day, alliinase in the presence of glycerol showed, on average, 4.5-times higher activity compared to glycerol-free Tricine-KOH buffer.

Addition of EDTA, metal chelator preventing metal-induced oxidation of thiol groups, resulted in lower initial activity (measured at the 0 h). Although, during the first three hours, the observed values were 2-times higher than those of the reference sample. After 24 hours, the enzyme showed lower activity than the sample with ascorbic acid. However, enzyme activity was still 4.7-times higher than the reference value.

Next, DTT (dithiothreitol) and its effect on the stabilization of the enzyme was investigated. DTT showed a significant, but short-lived positive effect for the first 3 hours, while the activity after 24 h was comparable to the reference sample.

Finally, the combined effects of the most beneficial additives and salts, i.e., ascorbic acid (4 mM) and NaCl (50 mM) on the enzyme activity were examined (Fig 6B). The reference sample without any additives showed a typical decrease in the enzyme activity, which can be described by first-order kinetics. On the other hand, the presence of NaCl or ascorbic acid showed non-monotonous time-dependence of enzyme activity with a peak at 2 h and significantly higher activity compared to the reference. When both ascorbic acid and NaCl were present at the same time, alliinase showed lower activity at 1 h followed by the highest enzyme activity recorded after 2, 3 and 6 hours, which was more than 8x higher than for the reference sample.

## The effect of alliin enantiomers

Synthetically prepared alliin ($M_w$ = 177.22 g/mol) exists as a mixture of two diastereomers, (2R)-2-amino-3-[(S)-prop-2-enylsulfinyl] propanoic acid (natural alliin), (2R)-2-amino-3-

[(R)-prop-2-enylsulfinyl] propanoic acid, because of the asymmetry of the sulfoxide group. Therefore, the maximal velocities $V_{max}$ concerning natural L-(+)-alliin and diastereomixture of L-alliin were investigated. The value of $V_{max}$ for the mixture of diastereomers at 25°C was in average 52% lower compared to the natural alliin, which is given by a higher selectivity of alliinase toward naturally occurring substrate [52, 53].

## Storage stability of lyophilized alliinase

The formulation into a solid dry product is considered the optimal strategy for preservation of the enzyme activity and to achieve an acceptable shelf life. It is well-known that the presence of moisture may have a serious impact on the stability of lyophilized proteins [54, 55]. However, to the best of our knowledge, the stability of lyophilized dry alliinase concerning temperature and various levels of relative humidity has not been yet explored. In order to study the adverse effect of water absorption, a dry lyophilized form of the enzyme was exposed to pre-set relative humidity (RH) levels, i.e., 0, 11, 50, 75 and 100% RH, and temperatures 25°C and 37°C, for one week, during which period the samples were periodically monitored for remaining enzyme activity. The corresponding values of equilibrium relative humidity are listed in Table 1. The results showed that one-week storage under inert nitrogen atmosphere (0% RH) at 25°C (half-life $t_{0.5} = 43$ days) and 37°C ($t_{0.5} = 32$ days) preserved a high activity (>70%), which is due to the absence of both oxygen and moisture. The observed drop in activity after the first day compared to the activity of the lyophilized powder stored at -20°C (reference value of 100%) is attributed to the moisture uptake and thermal/oxidative stress associated with sample heating and handling. Enzyme activity and $t_{0.5}$ were reduced significantly with the increase of RH and storage temperature, as is summarised in Fig 7A and 7B. For instance, increasing the RH value from 0 to 11, 50, 75 and 100% resulted in a reduction of $t_{0.5}$ from 43 to 11, 5, 4 and 2 days, respectively, measured at 25°C. For both temperatures holds that activity of the enzyme stored at X% RH is approx. 2-times higher than of the same enzyme stored at X +20% RH. Therefore, for the long-term storage, the purified alliinase in the lyophilized form should be kept at conditions reducing water activity (frozen state) or under an oxygen-free atmosphere and sufficiently low RH level.

## Antibacterial potency of the alliin-alliinase system

Antibacterial assay of produced allicin was studied by a disk diffusion method against *E. coli*, *P. putida* and *S. epidermidis* as inhibition growth zones emerging after 24h of incubation period in the vicinity of the application sites (Fig 8A). Glycosidic antibiotic kanamycin (ATB)

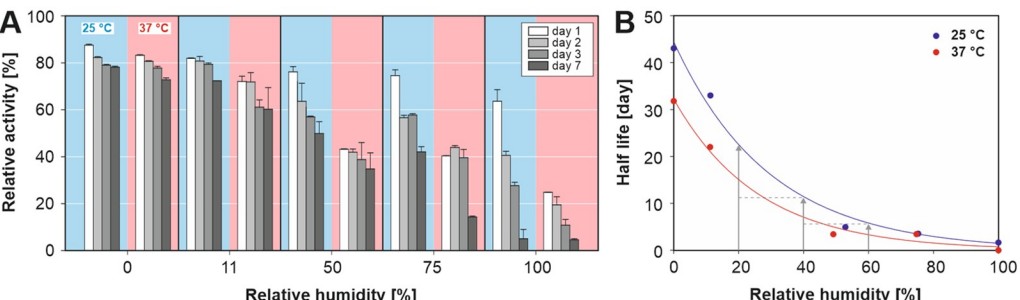

**Fig 7. A) Effect of storage conditions on the activity of lyophilized alliinase; B) calculated half-life of alliinase vs relative humidity and storage temperature.** The initial activity of the lyophilized powder stored at -20°C was taken as a reference value.

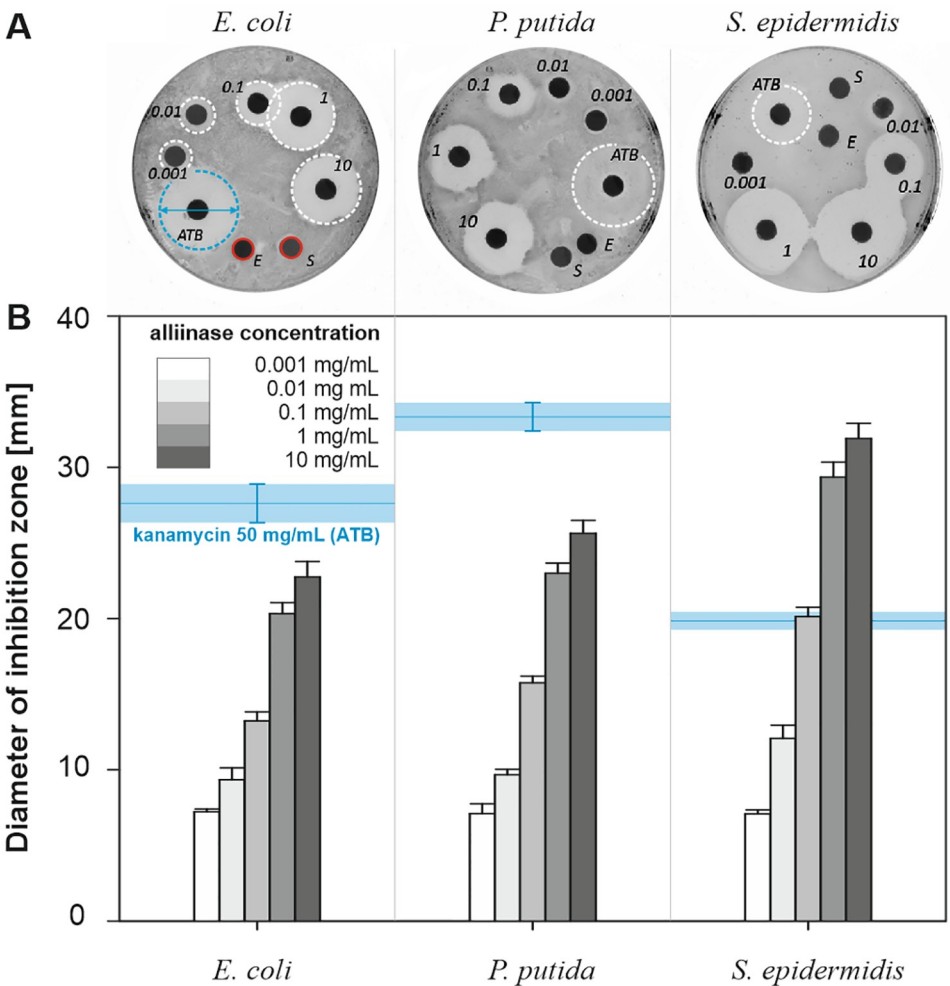

**Fig 8. A) Example of disk diffusion susceptibility testing of *E. coli*, *P. putida* and *S. epidermidis* (numbers correspond to alliinase concentration in mg/mL), ATB—kanamycin, E—enzyme, S—substrate; B) influence of alliinase concentration (0.001 to 10 mg/mL) in combination with 100 mM alliin on the diameter of inhibition zones.** Error bars represent standard deviation based on three independent experiments.

was used as a reference (50 mg/mL, 20 μL) for comparison of antibacterial effect for specific samples and bacterial strains [51]. Control samples, 100 mM alliin (S) and 5 mg/mL alliinase (E) applied separately, did not cause the formation of inhibition zones, i.e. no observable antibacterial effects. The results summarised for all studied bacterial strains and various alliinase amounts (from 0.001 to 10 mg/mL) in the presence of 100 mM alliin are shown in Fig 8B. Even the lowest tested concentration of alliinase (enzyme to substrate mass ratio = $5.6 \times 10^{-5}$) showed detectable inhibition zones for all studied bacterial strains. Higher enzyme content resulted in the formation of larger inhibition zones, which can be explained by the higher alliin conversion. Gram-positive *S. epidermidis* showed the largest inhibition zones with respect to allicin, followed by Gram-negative *E. coli* and *P. putida* (in that order).

A series of additional measurements with finer concentration steps was conducted to evaluate the contribution of both the enzyme and the substrate against *E. coli*, which represents bacterial strain standing in the middle of the susceptibility spectrum of studied bacteria. Similarly

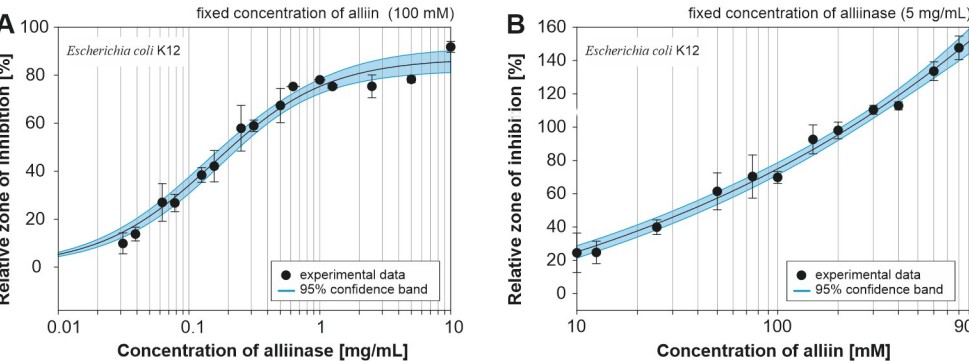

**Fig 9. Effect of alliinase (A) and alliin (B) concentration on the inhibition of _E. coli_.** The inhibition zone for kanamycin (50 mg/mL, 20 μL) served as a reference value in both cases; x-axis in logarithmic scale; error bars represent standard deviation based on three independent experiments.

to results shown in Fig 8, the fixed concentration of alliin (100 mM) was mixed with alliinase solution of concentration 0.03 to 10 mg/mL. The diameter of the inhibition zone for 50 mg/mL kanamycin was used as the reference value for comparison of observed antibacterial effects of the produced allicin. Fig 9A illustrates that increasing alliinase amount provided a larger zone of inhibition, which is in agreement with results presented in Fig 8. However, the application of alliinase concentration higher than 0.6 mg/mL did cause only marginal improvement of the inhibition of bacterial growth which indicates that most of the alliin was converted.

The influence of substrate concentration on the diameter of inhibition zones for _E. coli_ was investigated. Inhibition zones corresponding to a mixture of alliin (in the concentration range of 10 to 800 mM) and 5 mg/mL alliinase are summarized in Fig 9B. It was confirmed, that with a higher initial concentration of alliin, the diameter of observed inhibition zones increased.

## Bactericidal potency of _in-situ_ formed allicin

The Live/Dead cell assay based on the propidium iodide and fluorescein staining was used to study the time-resolved antibacterial effect of allicin. Fig 10A presents the time evolution of

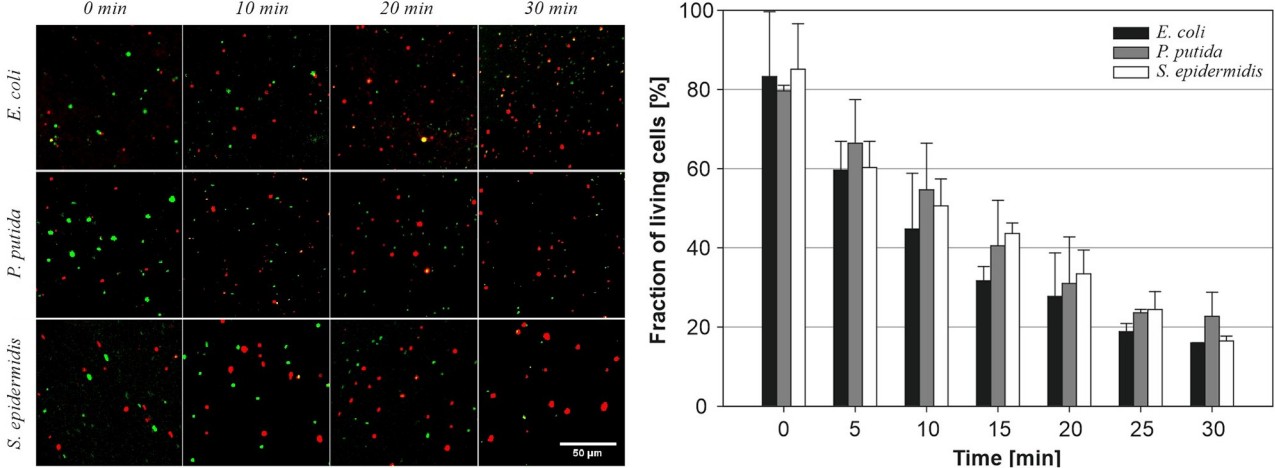

**Fig 10. Cell viability assay of bacterial suspension (300 μL) incubated for 0 to 30 min with a mixture of alliin (100 μL; 100 mM) and alliinase (100 μL; 1 mg/mL).** Ten fields or more were analyzed in triplicates; the scale bar corresponds to 50 μm; the total number of bacteria at a given time was taken as 100%.

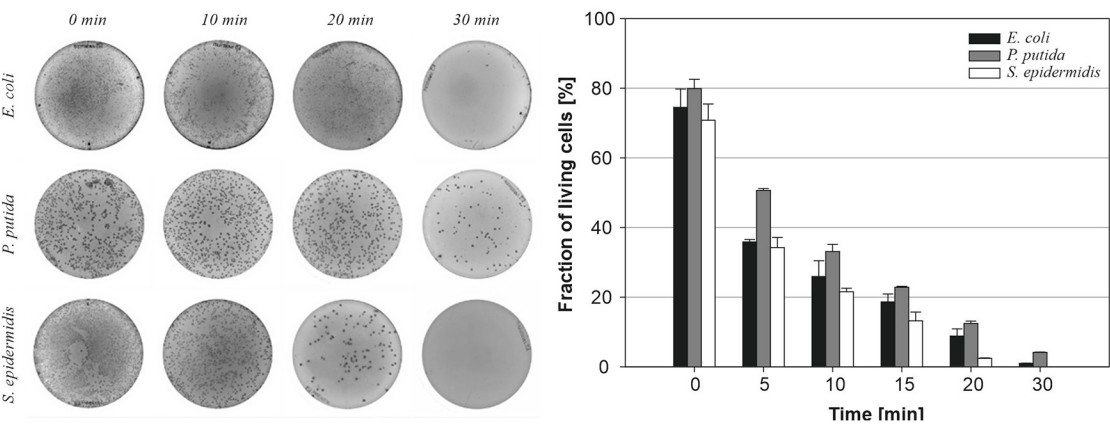

**Fig 11. Bactericidal effect of the allicin evaluated by the plate count method.** Left—agar plates with bacterial colonies after plating of the sample; Right—quantification of plate count method. The total count of emerging colonies from the untreated sample was taken as 100%.

bacterial viability immediately upon allicin addition (0 min), after 10, 20 and 30 min. The negative control bacterial samples without allicin, i.e., without alliin or enzyme, remained highly viable over 30 min. When both the enzyme and the substrate were present in the bacterial sample (enzyme—0.2 mg/mL, alliin—20 mM), the fraction of living cells exposed to allicin dropped after 30 min to 16% (*E. coli*, *S. epidermis*) and 23% (*P. putida*). The cell viability half-life (based on first-order kinetics) for *E. coli*, *P. putida* and *S. epidermidis*, was calculated as 12, 15 and 14 min, respectively.

The allicin bactericidal effect assay was verified in parallel by plate count method of viable bacteria (Fig 11). The results of the plate count method were in agreement with Live/Dead assay, and it was revealed that alliin/alliinase had the strongest bactericidal effect on *S. epidermidis*, followed by *E. coli* and *P. putida*, which is in agreement with results summarised in Fig 8.

### Allicin-induced cell damage

The impact of allicin on the morphological changes of *E. coli*, *P. putida* and *S. epidermidis* was investigated using SEM analysis (Fig 12). Gram-positive *S. epidermidis* changed morphology immediately upon allicin formation, which was indicated by the presence of cell fragments and intracellular matter enveloping a cluster of cells with typical cocci shape. After 30 min, no intact cells, and only cell fragments were observed. In contrast to *S. epidermidis*, Gram-negative *P. putida* presented small, but apparent morphological changes upon the addition of allicin (0 min), with wrinkles and small bumps observed on the cell body. After 30 min, most cells were disrupted, but some intact cells were still visible. Finally, *E. coli* showed no visible morphological changes immediately after allicin addition. After 30 min, some disrupted cells were found, but compared to the other two bacterial strains, the allicin-induced damage was relatively small. The results show clearly the difference in the dynamic of allicin action against Gram-positive and Gram-negative bacterial strains.

### Conclusion

In this comprehensive study, the isolated influence of numerous factors on operational and storage stability of purified garlic alliinase is presented and discussed. It was demonstrated that

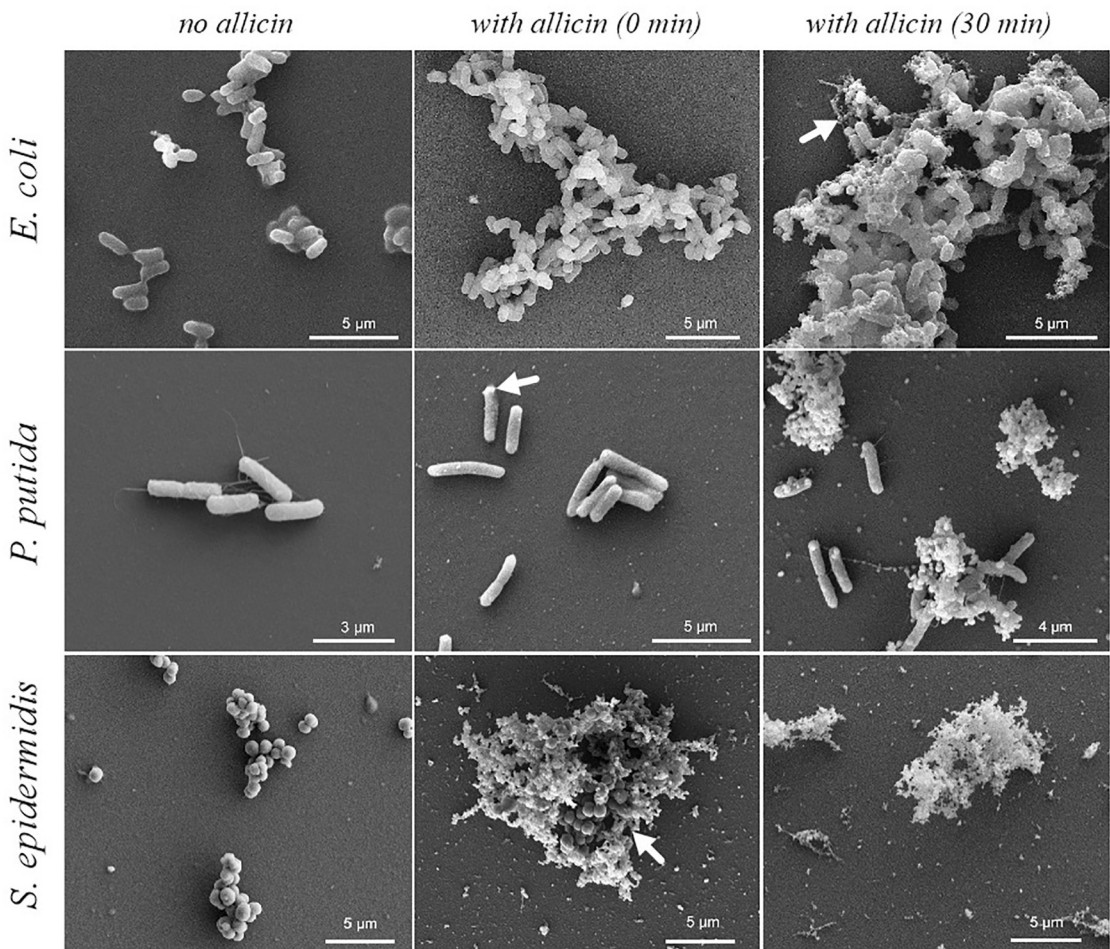

**Fig 12. Cell morphology of *E. coli*, *P. putida* and *S. epidermidis* before (no allicin—The left column), right after (time 0 min—Middle column) and after 30 min upon allicin addition (the right column); Arrows show the first appearance of morphological changes.**

alliinase incubated in Tricine-KOH or PBS buffers, instead of phosphate buffers generally used in alliinase-alliin related studies, showed improvement in both short-term stability and activity. However, alliinase incubated in pure Tricine-KOH buffer was stabilized only for several hours. Therefore, the effect of chloride salts (NaCl, KCl, CaCl$_2$ or MgCl$_2$) on the stability of alliinase solution stability was explored. The comparison showed that NaCl had among studied salts the most noticeable stabilizing effect. In addition to salts, the contribution of other additives (antioxidants, osmolytes, chelating and reducing agents) was verified, from which ascorbic acid showed a superior stabilizing effect. The simultaneous presence of both NaCl and ascorbic acid stabilized alliinase in the solution for 6 hours at 25˚C, which in the context of non-immobilized alliinase represents a considerable improvement, and an eight-times higher activity compared to Tricine-KOH buffer alone. The storage stability of lyophilized alliinase was examined for predefined relative humidity (RH) levels and storage temperatures (25˚C and 37˚C). It was demonstrated that with every 20% of RH, alliinase activity was reduced by 50% at specified times. Finally, alliinase activity and *in-situ* formation of antibacterial allicin were monitored by static disk diffusion method and dynamic viability assays showing the

progress of allicin-induced cell damage on selected Gram-positive and Gram-negative bacterial strains.

Thorough knowledge regarding the time-dependent response of purified alliinase in the non-immobilized form to various factors, i.e. pH, temperature, type of buffer, and the presence of a variety of additives, significantly contribute to the development of formulations containing stabilized alliinase for the synthesis of allicin or its chemical analogues. The potential use of environmentally friendly and time-proved *Allium* chemistry in human or veterinary medicine (treatment of bacterial infections, reduction of blood pressure) or agriculture (plant protection, urease inhibitors in fertilizers) is great, and we believe, that the findings presented here will contribute towards the development of nature-based products harnessing the volatile nature of highly potent allicin.

## Supporting information

**S1 Raw images.**
(TIF)

## Author Contributions

**Conceptualization:** Zdeněk Knejzlík, Viola Tokárová, Ondřej Kašpar.

**Data curation:** Petra Janská, Ayyappasamy Sudalaiyadum Perumal, Radek Jurok.

**Formal analysis:** Petra Janská, Zdeněk Knejzlík, Ayyappasamy Sudalaiyadum Perumal, Ondřej Kašpar.

**Funding acquisition:** Petra Janská, Viola Tokárová, Dan V. Nicolau.

**Investigation:** Petra Janská, Zdeněk Knejzlík.

**Methodology:** Ayyappasamy Sudalaiyadum Perumal.

**Project administration:** Viola Tokárová.

**Resources:** Viola Tokárová, Dan V. Nicolau, František Štěpánek.

**Supervision:** Zdeněk Knejzlík, Viola Tokárová, Dan V. Nicolau, František Štěpánek, Ondřej Kašpar.

**Visualization:** Ondřej Kašpar.

**Writing – original draft:** Petra Janská, Viola Tokárová, Ondřej Kašpar.

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
