## [Decision Letter · Decision Letter 0]

16 Nov 2020

PONE-D-20-24290

Effect of physicochemical parameters on the stability and activity of garlic alliinase and its use for in situ allicin synthesis

PLOS ONE

Dear Dr. Kašpar,

Thank you for submitting your manuscript to PLOS ONE. After careful consideration, we feel that it has merit but does not fully meet PLOS ONE’s publication criteria as it currently stands. Therefore, we invite you to submit a revised version of the manuscript that addresses the points raised during the review process.

We look forward to receiving your revised manuscript.

Kind regards,

Pradeep Kumar

Academic Editor

PLOS ONE

Journal Requirements:

3. Please note that PLOS does not permit references to “data not shown.”

Authors should provide the relevant data within the manuscript, the Supporting Information files, or in a public repository.

If the data are not a core part of the research study being presented, we ask that authors remove any references to these data.

4. Please provide the source (name and location of company, market) of the garlic used in the study.

5. To comply with PLOS ONE submission guidelines, in your Methods section, please provide additional information regarding your statistical analyses.

For more information on PLOS ONE's expectations for statistical reporting, please see https://journals.plos.org/plosone/s/submission-guidelines.#loc-statistical-reporting

6. At this time, we ask that you please provide scale bars on the microscopy images presented in Figure 10 and refer to the scale bar in the corresponding Figure legend.

Reviewers' comments:

Reviewer's Responses to Questions

**Comments to the Author**

1. Is the manuscript technically sound, and do the data support the conclusions?

Reviewer #1: Partly

Reviewer #2: Yes

Reviewer #3: Yes

2. Has the statistical analysis been performed appropriately and rigorously? 

Reviewer #1: No

Reviewer #2: I Don't Know

Reviewer #3: No

3. Have the authors made all data underlying the findings in their manuscript fully available?

Reviewer #1: Yes

Reviewer #2: Yes

Reviewer #3: Yes

4. Is the manuscript presented in an intelligible fashion and written in standard English?

Reviewer #1: Yes

Reviewer #2: Yes

Reviewer #3: Yes

5. Review Comments to the Author

Reviewer #1: Dear Authors,

Green chemistry, also called sustainable chemistry, is a rapidly growing field of chemistry that is based on the principles of reducing the impact of chemical production on the environment and human health. Your article “Effect of physicochemical parameters on the stability and activity of garlic alliinase and its use for in situ allicin synthesis” and idea of using allinase to form in situ the bactericidal agent allicin is consistent with the principles of green chemistry. The manuscript is well written and understandable. In my opinion, the article fully complies with the PLOS ONE editorial criteria. Much attention is paid to the technical aspects of the effects of various physicochemical parameters on allinase activity. This work may have an impact on the relevant field of science.

However, below you can find my minor suggestions for improving the manuscript.

1) The main limitation of the work is that you use a system of coupled reactions to analyze the effect of physicochemical parameters on allinase activity. In the first stage of the reaction, pyruvic acid is formed, which is then reduced by LDH in the presence of NADH. The analyzed physicochemical parameters can affect the activity of the second enzyme (LDH). Thus, the reported effects, for example of additives and pH, can be related to the effect on both alliinase and the second enzyme. However, LDH is known to be a stable enzyme. I believe this limitation should be noted in the discussion. It is necessary to indicate works that confirm the stability of this enzyme.

2) I was surprised by the lack of a “Statistical Analysis” section. Please add this section and include the following information. How is the data presented (Mean ± SD?)? What statistical criteria were used to compare different groups?

3) Lines 111-112. The characterization of the substrate used is very important for biochemical research. Please indicate the synthesis method (as well as the reference), describe the purity of the obtained alliin. Please interpret the abbreviation “UCT”.

4) Lines 133-135. In the sentence “The characterization of the enzyme was carried out by using sodium dodecyl sulfate polyacrylamide gel electrophoresis (SDS-PAGE) run under reducing conditions (in the presence of β-ME)” the word “run” may be redundant.

5) “1.4 Enzyme assay”. I assume that you used the previously developed method for determining of alliinase activity. Please provide a reference to the method. Could this be the method of Selby and Collin 1976? Lines 150-153: Indicate the concentration of alliinase used. Please indicate how the Michaelis – Menten kinetic constants are calculated. Which program was used? Have you used the non-linear approximation by the least square method in the Origin program?

6) Lines 265-266. Please indicate the concentration of the alliinase. Could you explain the choice of this enzyme concentration? I think you know that for the Michaelis - Menten kinetic constants analysis it is necessary to use the enzyme concentration corresponding to the linear region of the dependence of the initial reaction rate on the enzyme concentration. Have you conducted such experiments? It is necessary to provide this data.

The Vmax value depends on the enzyme concentration. Please calculate kcat values and compare with previously known data for alliinase.

7) Figure 4 A. Please indicate in the caption to the figure, what was taken as 100%? In Figure 5, you indicated that.

8) Line 367. “statistically relevant relationship” Please indicate what you mean. What statistical criterion was used?

9) Figure 6 A,B. What is the initial reference value? Please indicate in the caption to the figure or in the text, what was taken as 100%? I may be wrong, but you take five initial reference values for each of the five value pairs (0, 1, 2, 3, 24h). I propose to recalculate the data and take the initial reference value of the activity in 0 hours of incubation with no additives in all cases. Then the graph will be clearer.

10) Figure 7 A. Please indicate in the caption to the figure or in the text, what was taken as 100%?

11) Figure 8. “Note: …” This sentence can be transferred to the methods, or you can add a similar “Note:…” to each figure, if applicable.

12) Figure 9 A: Indicate that the measurements were taken at a fixed concentration of alliin. Figure 9 B: Indicate that measurements were taken at a fixed enzyme concentration. Please indicate the appropriate concentration in the text or caption.

13) My comment on lines 466-468. The absence of a plateau may be associated with a low concentration of alliinase, or its partial destruction by bacterial proteases. If you analyzed the dependence of the initial reaction rate on the enzyme concentration (see my comment 6), this would be more understandable.

14) Figure 10 and 11. On the right side of the pictures, what was taken as 100%?

You have done a great and interesting work.

Good luck with your further experiments.

Best regards

Reviewer #2: Manuscript evaluation: PONE-D-20-24290

Effect of physicochemical parameters on the stability and activity of garlic alliinase and its use for in situ allicin synthesis

General Comment

In this manuscript, authors described studies about the effect of physicochemical parameters on the stability and activity of garlic alliinase and its use for in situ allicin synthesis. Authors tried to provides a comprehensive insight into the alliinase stability and activity in the solution and lyophilized state as well as the influence of commonly used buffers, and some additives, on the enzyme activity and stability. While the manuscript contains massive data and results, only some interesting results, mainly the effect of additives on alliinase stability and Storage stability of lyophilized literature could be considered as novel. All others data were vastly reported in the literature. Hence I consider that the manuscript is not sufficiently original and should be rejected at its actual form. If the authors could identify and strength studies and results not previously reported, manuscript could be reconsidered for publication.

Reviewer #3: 1. The manuscript needs revision with respect to the typos, grammar and language

2. 1.1 Chemicals - This section needs to be precise. Name of all chemicals not needed

3. Line 119. T Mallika to be written as Mallika

4. Line 125 Cheesecloth- cheese cloth

5. Minutes to be written as min

6. Line 136 give details of molecular mass markers used

7. BSA- Expand

8. 150. Maximum reaction rate - revise at maximum velocity.

9. Line 161 and 164 Temperature range is confusing. at Line 161 it 20 C and at Line 164 it is -20 C

10. room temperature (25 C) only mention 25 C

11. Give the details of lyophilization

12. Line 194. Table 25 and 37 C- write as 25 C and 37 C

13. 1.6. What is the source of these microorganisms? were these human pathogens?

14. Statistical analysis is missing

6. PLOS authors have the option to publish the peer review history of their article (what does this mean?). If published, this will include your full peer review and any attached files.

Reviewer #1: **Yes: **Evgeny Ermakov

Reviewer #2: No

Reviewer #3: **Yes: **R Z Sayyed

---

## [Author Response · Author response to Decision Letter 0]

22 Dec 2020

Response to the reviewers

All text changes in the manuscript are highlighted (blue colour). All image files were updated and uploaded.

Reviewer #1

We are very thankful to the reviewer for his time to read our manuscript thoroughly, for constructive comments helping us to improve the final form of the paper and for the appreciation of our work. We made appropriate changes in the manuscript to address the reviewer's questions and suggestions.

1) The main limitation of the work is that you use a system of coupled reactions to analyze the effect of physicochemical parameters on allinase activity. In the first stage of the reaction, pyruvic acid is formed, which is then reduced by LDH in the presence of NADH. The analyzed physicochemical parameters can affect the activity of the second enzyme (LDH). Thus, the reported effects, for example of additives and pH, can be related to the effect on both alliinase and the second enzyme. However, LDH is known to be a stable enzyme. I believe this limitation should be noted in the discussion. It is necessary to indicate works that confirm the stability of this enzyme.

We are thankful for this remark. We are aware that deactivation or significant decrease of LDH activity resulting in a comparable rate of pyruvate formation and conversion can lead to a misleading report of alliinase activity. Considering this fact, LDH was used in such excess that the conversion of alliin was at all times the rate-limiting step of the coupled reaction. To verify this statement and to show the dependence of LDH activity on temperature, pH and buffer type, additional experiments were performed. Enzyme assay using LDH and NADH (same amounts as stated in the manuscript) with added pyruvate (corresponding to the amount of alliin, 20 mM) proved that conversion of pyruvate to lactate was almost instant upon pyruvate addition almost invariant to examined conditions. Therefore, in order to determine LDH activity (based on time dependant UV-VIS measurements), the concentration was reduced 50-times. Experimental data (see below) show that LDH remained highly active at temperatures 25 °C to 42 °C and pH 5 to 8. 

Fig. 1 A) Effect of temperature, B) pH (PB buffers) and C) buffer type on the activity of LDH. LDH activity in Tric-KOH measured at 25 °C was used as a reference value (100 %).

For that reason, we concluded that LDH could be considered as sufficiently stable in the examined range of conditions which is in agreement with the literature (Jackson, E., et al. "Enhanced stability of l-lactate dehydrogenase through immobilization engineering." Process Biochemistry 51.9 (2016): 1248-1255). However, we observed a noticeable decrease in LDH activity and instability of NADH bellow pH 5. Therefore, a paragraph in Effect of pH, temperature and buffer on alliinase stability and activity was extended to discuss limitation related to LDH coupled reaction system. In the case of additives (salts, antioxidants, etc.), alliinase was incubated without LDH, which was present only when checking aliquots for remaining enzyme activity at specific times. 

2) I was surprised by the lack of a "Statistical Analysis" section. Please add this section and include the following information. How is the data presented (Mean ± SD?)? What statistical criteria were used to compare different groups? 

The new section "Statistical evaluation" was added to the manuscript. Enzyme activities were expressed as means of 6 or more measurements and their respective standard deviation (SD). Data were analyzed using one-way ANOVA followed by Tukey and Dunnett tests to compare the influence of particular additives on enzyme activity. Values of P < 0.01 were considered to be statistically significant. Statistical analyses were performed using software SigmaPlot 11.

3) Lines 111-112. The characterization of the substrate used is very important for biochemical research. Please indicate the synthesis method (as well as the reference), describe the purity of the obtained alliin. Please interpret the abbreviation "UCT". 

A new paragraph describing alliin synthesis was added to Materials and methods section. References and information about the purity of synthetic alliin were included in the main text. The abbreviation "UCT" (University of Chemistry and Technology Prague) was removed. 

4) Lines 133-135. In the sentence "The characterization of the enzyme was carried out by using sodium dodecyl sulfate polyacrylamide gel electrophoresis (SDS-PAGE) run under reducing conditions (in the presence of β-ME)" the word "run" may be redundant.

The word "run "was removed from the text, as suggested.

5) "1.4 Enzyme assay". I assume that you used the previously developed method for determining of alliinase activity. Please provide a reference to the method. Could this be the method of Selby and Collin 1976? Lines 150-153: Indicate the concentration of alliinase used. Please indicate how the Michaelis – Menten kinetic constants are calculated. Which program was used? Have you used the non-linear approximation by the least square method in the Origin program?

For the enzyme assay, we followed the procedure by Jansen, H., B. Müller, and K. Knobloch, Characterization of an Alliin Lyase Preparation from Garlic (Allium sativum).Planta Med, 1989. 55(05): p. 434-439.. The reference [30] was added to the manuscript. 

The concentration of enzyme is now included in the caption of the Fig. 2B. Michaelis – Menten equation was applied to calculate the kinetic constant using SigmaPlot 11 software by non-linear least square regression. Details about the evaluation of kinetic constants were added to the main text. 

 

6) Lines 265-266. Please indicate the concentration of the alliinase. Could you explain the choice of this enzyme concentration? I think you know that for the Michaelis - Menten kinetic constants analysis it is necessary to use the enzyme concentration corresponding to the linear region of the dependence of the initial reaction rate on the enzyme concentration. Have you conducted such experiments? It is necessary to provide this data. The Vmax value depends on the enzyme concentration. Please calculate kcat values and compare with previously known data for alliinase.

Information about alliinase concentration was added to the caption of Fig. 2B. For simplicity, the value Vmax was recalculated to Vmax per mg of pure protein. The turnover number kcat for alliinase was calculated to be 193 s-1 which is comparable with available reports, e.g. kcat = 180 s-1 (Musah, Rabi A., et al. "Studies of a novel cysteine sulfoxide lyase from Petiveria alliacea: the first heteromeric alliinase." Plant physiology 151.3 (2009): 1304-1316). 

A suitable concentration of alliinase was optimized beforehand, as illustrated in Fig. 2 (bellow). Alliinase concentration of 5 µg.ml-1 was used for the alliinase kinetic characterization while 25 µg.ml-1 was used for all the other experiments. In both cases, alliinase concentration was within the linear range of the dependence of the initial reaction rate on the enzyme concentration as indicated by arrows and solid line in Fig. 2.

Fig. 2 Dependence of the initial reaction velocity (s-1) versus alliinase concentration at 25 °C; alliin concentration was 20 mM.

7) Figure 4 A. Please indicate in the caption to the figure, what was taken as 100%? In Figure 5, you indicated that.

The caption of Figure 4A was updated.

 

8) Line 367. "statistically relevant relationship" Please indicate what you mean. What statistical criterion was used?

The sentence with expression "statistically relevant relationship" was changed for clarification: 

"However, no relationship between the overall glycerol content and observed alliinase activity has been found."

9) Figure 6 A,B. What is the initial reference value? Please indicate in the caption to the figure or in the text, what was taken as 100%? I may be wrong, but you take five initial reference values for each of the five value pairs (0, 1, 2, 3, 24h). I propose to recalculate the data and take the initial reference value of the activity in 0 hours of incubation with no additives in all cases. Then the graph will be clearer.

The note about reference value (100 %) was added to the caption of Figure 6 A, B. 

The alliinase activity (Tric-KOH, no additives, 0 h) was taken as a reference (100 %) in all cases. There was only one reference value for each figure. 

10) Figure 7 A. Please indicate in the caption to the figure or in the text, what was taken as 100%?

The note about reference value was added to the caption of Fig. 7 A. 

11) Figure 8. "Note: …" This sentence can be transferred to the methods, or you can add a similar "Note:…" to each figure, if applicable.

All figure captions were revised.

12) Figure 9 A: Indicate that the measurements were taken at a fixed concentration of alliin. Figure 9 B: Indicate that measurements were taken at a fixed enzyme concentration. Please indicate the appropriate concentration in the text or caption.

The information about the fixed concentration of alliin and alliinase was added to Figure 9 A and B. 

9 A: calliin = 100 mM, 9 B: calliinase = 5 mg/ml

13) My comment on lines 466-468. The absence of a plateau may be associated with a low concentration of alliinase, or its partial destruction by bacterial proteases. If you analyzed the dependence of the initial reaction rate on the enzyme concentration (see my comment 6), this would be more understandable.

We agree that depicted reasons could be linked with the absence of the plateau. The experimental system is rather complex, and therefore, we changed the sentence as follows: 

"It was confirmed, that with a higher initial concentration of alliin, the diameter of observed inhibition zones increased."

14) Figure 10 and 11. On the right side of the pictures, what was taken as 100%?

The following text was added to the caption of Fig. 10 and 11. 

Figure 10 - the total number of bacteria at a given time (alive and death) was taken as 100 %. 

Figure 11 - the total count of emerging colonies of the untreated sample was taken as 100 %.

Reviewer #2 

In this manuscript, authors described studies about the effect of physicochemical parameters on the stability and activity of garlic alliinase and its use for in situ allicin synthesis. Authors tried to provides a comprehensive insight into the alliinase stability and activity in the solution and lyophilized state as well as the influence of commonly used buffers, and some additives, on the enzyme activity and stability. While the manuscript contains massive data and results, only some interesting results, mainly the effect of additives on alliinase stability and Storage stability of lyophilized literature could be considered as novel. All others data were vastly reported in the literature. Hence I consider that the manuscript is not sufficiently original and should be rejected at its actual form. If the authors could identify and strength studies and results not previously reported, manuscript could be reconsidered for publication.

We are grateful to the reviewer for his/her time and effort to read our work. However, we believe that reported results in terms of alliinase response to various stimuli goes beyond previously published findings and significantly contribute to the current knowledge in the field of natural antibiotics based on Allium chemistry. We are aware there is an overlap with existing literature regarding the influence of pH, temperature and presence of various cations on alliinase activity. It should be emphasized, that articles published to this date regarding alliinase use various enzyme sources, different isolation/purification protocols and assays which may significantly affect overall enzyme behaviour and makes direct comparison among them difficult, if even possible. In some cases, there is a significant contradiction among reported results regarding the effect of particular ions (e.g. Na+ and K+ cations). Most of the alliinase-related reports put the main emphasis on the study of alliinase activity, whereas alliinase stability is investigated sporadically. For all these reasons, we decided to explore the influence of commonly used buffers, salts and other additives/or their combinations on alliinase activity and time-dependent stability in the extend never conducted before. 

Another significant part of this work is dedicated to the study of lyophilized alliinase and its response to environmental conditions mimicking storage and the intended site of application (temperature, humidity, oxygen presence). The lack of any information about alliinase stability in dry state motivated us to pursue this research critical for the development of nature-inspired antibacterial products, e.g. powders and healing patches, relying on in-situ production of highly potent allicin. 

Finally, the third part of the proposed manuscript is dedicated to antibacterial testing going beyond commonly used disc diffusion method. In this work, we are providing insight into dynamics of allicin's action against selected bacterial strains using confocal microscopy, plate count method and visual observation of allicin induced damage of bacterial cell at specific times upon allicin addition.

Reviewer #3

We want to thank the reviewer for his time to read our work and for all comments and suggestions, improving the quality of the final text. All 14 comments were addressed, and appropriate changes have been done in the manuscript.

1. The manuscript needs revision with respect to the typos, grammar and language - 

 The main text has been revised and checked for grammatical errors. 

2. 1.1 Chemicals - This section needs to be precise. Name of all chemicals not needed

Generally used chemicals as solvents were omitted from "Chemicals" section.

3. Line 119. T Mallika to be written as Mallika

The reference "T. Mallika" was changed to Mallika. 

4. Line 125 Cheesecloth- cheese cloth

The term cheese cloth is used in the manuscript.

5. Minutes to be written as min

The term "minutes" was changed to min in the whole manuscript (including figures).

6. Line 136 give details of molecular mass markers used 

 The details about used molecular mass markers were added to the manuscript.

7. BSA- Expand

 Information about BSA concentration was added to the main text.

8. 150. Maximum reaction rate - revise at maximum velocity. 

The term "maximum reaction rate" was changed to maximum velocity.

9. Line 161 and 164 Temperature range is confusing. at Line 161 it 20 C and at Line 164 it is -20 C 

 The text was referring to the activity and stability of alliinase. For better understanding, the original text was revised and divided into two separate paragraphs. The first paragraph refers to alliinase activity measured at 20 °C to 42 °C, whereas the second refers only to the long-term stability of alliinase stored under different conditions (-20 °C to 37 °C).

10. room temperature (25 C) only mention 25 C

The term room temperature was changed to 25 °C everywhere in the manuscript.

11. Give the details of lyophilization 

 A detailed description of the lyophilization process was added to Section - Extraction and purification of alliinase.

12. Line 194. Table 25 and 37 C- write as 25 C and 37 C

The caption of the table was changed to "Conditions for testing of enzyme stability at 25 °C and 37 °C."

13. 1.6. What is the source of these microorganisms? were these human pathogens? 

The source of bacterial strains with the description of biosafety level was added to Antibacterial assays section. Escherichia coli strain K12 is not human pathogen, and it is widely used as wild type model microorganism. Staphylococcus epidermidis is a common component of animal (including human) microflora. However, it can cause opportunistic nosocomial infections in immunodeficient persons. It is also bacterial species forming a biofilm on the surgical devices such as joint replacements. Pseudomonas putida is environmental bacterial species living in the soil and water. However, some strains can cause severe infections in humans.

14. Statistical analysis is missing 

The section Statistical evaluation was added to the manuscript.

Enzyme activities were expressed as means of 6 or more measurements and their respective standard deviation (SD). Data were analyzed using one-way ANOVA followed by Tukey and Dunnett tests to compare the influence of particular additives on enzyme activity. Values of P < 0.01 were considered to be statistically significant. Statistical analyses were performed using software SigmaPlot 11.

 

All journal requirements were addressed and resolved in full extend.

Journal Requirements:

The manuscript template was edited to meet PLOS ONE style requirements. 

2. PLOS ONE now requires that authors provide the original uncropped and unadjusted images underlying all blot or gel results reported in a submission's figures or Supporting Information files. This policy and the journal's other requirements for blot/gel reporting and figure preparation are described in detail at https://journals.plos.org/plosone/s/figures#loc-blot-and-gel-reporting-requirements and https://journals.plos.org/plosone/s/figures#loc-preparing-figures-from-image-files. When you submit your revised manuscript, please ensure that your figures adhere fully to these guidelines and provide the original underlying images for all blot or gel data reported in your submission. See the following link for instructions on providing the original image data: https://journals.plos.org/plosone/s/figures#loc-original-images-for-blots-and-gels.

The unedited raw version of image Fig.2B was uploaded as an individual file S1_raw_images. 

The same version of the image was uploaded to publicly-available data repository figshare.com 

• https://doi.org/10.6084/m9.figshare.13469238.v1

• 10.6084/m9.figshare.13469238

 3. Please note that PLOS does not permit references to "data not shown."

Authors should provide the relevant data within the manuscript, the Supporting Information files, or in a public repository.

If the data are not a core part of the research study being presented, we ask that authors remove any references to these data.

"Data not shown" was omitted in the main text.

 

4. Please provide the source (name and location of company, market) of the garlic used in the study.

Source of garlic was added to the main text in Materials and methods section: "Garlic was purchased from Farmers' Market, Zahradnictví Riegelová, country of origin Czech Republic." 

5. To comply with PLOS ONE submission guidelines, in your Methods section, please provide additional information regarding your statistical analyses.

For more information on PLOS ONE's expectations for statistical reporting, please see https://journals.plos.org/plosone/s/submission-guidelines.#loc-statistical-reporting

Statistical evaluation section was added to the main text. 

6. At this time, we ask that you please provide scale bars on the microscopy images presented in Figure 10 and refer to the scale bar in the corresponding Figure legend.

The scalebar was added to Fig. 10.

---

## [Decision Letter · Decision Letter 1]

8 Mar 2021

Effect of physicochemical parameters on the stability and activity of garlic alliinase and its use for in situ allicin synthesis

PONE-D-20-24290R1

Dear Dr. Ondřej Kašpar,

We’re pleased to inform you that your manuscript has been judged scientifically suitable for publication and will be formally accepted for publication once it meets all outstanding technical requirements.

Kind regards,

Pradeep Kumar

Academic Editor

PLOS ONE

Additional Editor Comments (optional):

The decision of manuscript acceptance based on two positive response from the reviewer. 

Reviewers' comments:

Reviewer's Responses to Questions

**Comments to the Author**

1. If the authors have adequately addressed your comments raised in a previous round of review and you feel that this manuscript is now acceptable for publication, you may indicate that here to bypass the “Comments to the Author” section, enter your conflict of interest statement in the “Confidential to Editor” section, and submit your "Accept" recommendation.

Reviewer #1: All comments have been addressed

Reviewer #2: (No Response)

Reviewer #3: All comments have been addressed

2. Is the manuscript technically sound, and do the data support the conclusions?

Reviewer #1: Yes

Reviewer #2: Partly

Reviewer #3: Yes

3. Has the statistical analysis been performed appropriately and rigorously? 

Reviewer #1: Yes

Reviewer #2: I Don't Know

Reviewer #3: Yes

4. Have the authors made all data underlying the findings in their manuscript fully available?

Reviewer #1: Yes

Reviewer #2: Yes

Reviewer #3: Yes

5. Is the manuscript presented in an intelligible fashion and written in standard English?

Reviewer #1: Yes

Reviewer #2: Yes

Reviewer #3: Yes

6. Review Comments to the Author

Reviewer #1: Dear Authors,

It seems to me that in this form your manuscript can be approved for publication in Plos One. In my opinion, the article fully complies with the PLOS ONE editorial criteria. After revision, the article became more understandable and readable. I am satisfied with the responses of the authors. I have no objection to the manuscript.

Good luck with your further experiments.

Best regards

Reviewer #2: While the revised manuscript seems to be enhanced according to the comments of others reviewers, I did not really see changes relating to my initial comment. Indeed I had advised to authors to strength studies and results not previously reported. Although I respect the opinion of others reviewer, but in line of my first opinion, my decision is reject.

Reviewer #3: (No Response)

7. PLOS authors have the option to publish the peer review history of their article (what does this mean?). If published, this will include your full peer review and any attached files.

Reviewer #1: **Yes: **Evgeny Ermakov

Reviewer #2: No

Reviewer #3: **Yes: **R. Z. Sayyed

---

## [Editor Report · Acceptance letter]

11 Mar 2021

PONE-D-20-24290R1 

Effect of physicochemical parameters on the stability and activity of garlic alliinase and its use for *in situ* allicin synthesis 

Dear Dr. Kašpar:

I'm pleased to inform you that your manuscript has been deemed suitable for publication in PLOS ONE. Congratulations! Your manuscript is now with our production department. 

Kind regards, 

on behalf of

Dr. Pradeep Kumar 

Academic Editor

PLOS ONE